# Indoor climbing and well-being of young adults: Perspectives among indoor climbers

Elsa C. Osborne[1], Jeff Rose [2], Logan Reeves[3], Kelli Spear[1], Max E. Coleman[4], Timothy A. Brusseau Jr[5], Kathy Franchek-Roa[6], Akiko Kamimura [4]*

1 Health, Society & Policy Program, College of Social & Behavioral Science, The University of Utah, Salt Lake City, Utah, United States of America, 2 Department of Parks, Recreation, & Tourism, College of Health, The University of Utah, Salt Lake City, Utah, United States of America, 3 Department of Biology, College of Science, The University of Utah, Salt Lake City, Utah, United States of America, 4 Department of Sociology, College of Social & Behavioral Science, The University of Utah, Salt Lake City, Utah, United States of America, 5 Department of Health and Kinesiology, College of Health, The University of Utah, Salt Lake City, Utah, United States of America, 6 Department of Pediatrics, School of Medicine, The University of Utah, Salt Lake City, Utah, United States of America

* akiko.kamimura@utah.edu, kamimura@umich.edu

## Abstract

The ongoing mental health crisis among college students raises the necessity of further research on physical, mental, and social well-being; and the role that indoor climbing can play in fostering social connection and improving mental health while also increasing physical fitness. Indoor climbing has many health benefits. Increasing numbers of young adults are involved in indoor climbing; however, research on health and indoor climbing for young adults is lacking. The objective of this qualitative study was to describe indoor climbing and its relationship with the physical, mental, and social well-being of young adults based on perceptions among indoor climbers. Thirty indoor climbers age 18–25 were interviewed from October 2023 to December 2023. The interviews were transcribed and coded using thematic methods. Most participants agreed that indoor climbing positively affected all three areas of their well-being. Themes emerged showing that climbing facilitates a sense of connection, acts as a form of mindfulness, boosts overall mental health, and encourages healthier lifestyles. The results of this study suggest indoor climbing may be effective to promote health among college students and young adults.

## Introduction

Indoor climbing is a sport that entails climbing indoor, constructed vertical walls that are designed to simulate outdoor rock cliffs, and this activity is growing in popularity among young adults [1]. Three common types of indoor climbing include bouldering, top rope, and lead climbing. Bouldering is a form of climbing with no rope or harness on walls averaging 10 feet in height [2]. Top rope and lead climbing involve ropes,

**Data availability statement:** All relevant data are within the paper.

**Funding:** This research was supported by 1U4U Innovation Funding and the Undergraduate Research Opportunity Program at the University of Utah. The funders had no role in study design, data collection and analysis, decision to publish, or preparation of the manuscript.

**Competing interests:** The authors have declared that no competing interests exist.

harnesses, and a partner; the walls are much taller, averaging around fifty feet [3]. Lead climbing is similar to top rope climbing, but the climber clips their rope into each pre-fixed anchor as they climb up the wall [4].

Previous studies have examined the benefits of climbing on well-being. Climbers have increased body awareness, improving their spatial awareness or proprioception [5]. Climbing has also been found to improve muscle strength –[6] and have therapeutic benefits among children and young adults with mental health conditions [7]. The implementation of a youth development program using rock climbing produces higher teamwork skills through the partnership involved in climbing [8]. The experience of climbing brings athletes into a state of meditative awareness through movement and a sense of presence [9]. Climbing can also act as a form of exposure therapy (ET) that helps break avoidant thought patterns [10]. Exposure therapy has been shown to ease depression and treat PTSD (Post Traumatic Stress Disorder) in patients over fourteen years old [11] and could be a potential health benefit for college-aged young adults, who report having exceptionally high stress levels [12].

Despite these benefits and the increased popularity of climbing, research is limited regarding the intersections of indoor climbing, health, and the college student and young adult populations. Rates of mental health conditions among college students have doubled over the past decade [13], due in part to social media use [14]. In-person social interaction is especially important during this time. Social support and self-esteem are key factors to reduce mental distress among college students [15]. Thus, studying the impacts of indoor climbing on the well-being of young adults could lead to new findings in this realm. This study focused on indoor climbing instead of outdoor climbing because indoor climbing is often more accessible due to location, preparation needed, and skill level.

This study is built on social influence theory. Social influence explains how social interactions with other people affect opinions, beliefs, and behaviors [16]. Social interactions improve physical, mental and social health. Health promotion and prevention of health problems are especially important for young adults, as this is a time for life-long habit formation [17]. This period can impact the health trajectory across the lifespan [18]. Indoor climbing is potentially one of the physical activities to promote health of young adults due to its increased popularity among this population and the nature of the sport involving social interactions. Based on social influence theory, this study examined how indoor climbing enhances social networks, socialization, and social norms and as a result, promotes physical activity and health beyond the climbing wall [7].

The purpose of this study is to explore indoor climbing and physical, mental, and social well-being among college-aged young adults based on interviews with indoor climbers. By describing how indoor climbing can influence physical health, mental well-being, and social connectedness, this research aims to provide valuable insights into how indoor climbing can be integrated into health promotion programs.

## Methods

### Setting

This qualitative study based on interviews obtained ethical approval from the University of Utah's Institutional Review Board (IRB) (IRB# 00169118). Data were collected using a combination of in-person and virtual interviews via the video conference software ZOOM from October to December, 2023. The recruitment period for this study was from September 17, 2023 to December 2, 2023.

This project was carried out in the metropolitan area in the Intermountain West in a state conducive to living an active lifestyle such as hiking, skiing, climbing, backpacking, cycling, camping, and rafting [19]. Indoor climbing is a popular leisure activity in the state. The county where this study was conducted has nine indoor climbing gyms, including the university's climbing gym [20].

### Participants and data collection

Data were collected from October 2023 through December 2023. Participants were recruited through the distribution of flyers across the University of Utah campus and Salt Lake City's climbing gyms, social media posts, and snowball sampling. Snowball sampling was used because people in the climbing community tend to have social networks with other climbers. Non-probability sampling methods were used to recruit participants. Research assistants set up times to meet with participants for the interview. Consent was obtained from each participant. The written consent was waived by the IRB. The consent cover letter stated, "By participating in the focus group, you are giving your consent to participate." Documentation of informed consent be waived by the IRB (consent without signature) because "The research presents no more than minimal risk of harm to subjects and involves no procedures for which written consent is normally required outside of the research context." Eligibility criteria included being 18–25 years old and having participated in indoor climbing at least several times a month for the past six months or more. Thirty young adults participated in this study. Table 1 summarizes and describes the characteristics of participants: 46.67% of participants were male, 50% female, and 3.33% non-binary/transgender[1]. The majority of participants were white. Seventy percent of the participants were undergraduate students while 7% were graduate students. Twenty three percent were not currently in school.

Each participant was interviewed in-person at a local climbing gym (n=14) or via the videoconferencing application Zoom (n=16). Non-probability sampling methods were used to recruit participants, including snowball, convenience, and purposive sampling. Research assistants then set up times to meet with participants for the interview. The four research assistants were undergraduate students who received relevant training in research ethics and interviewing. Consent was obtained from each participant. Participants filled out the demographic survey prior to their interview. The interview was then conducted based on the interview guide. Participants received a $20 gift card when they completed an interview.

### Interview guide

The interview guide was developed based on the Seidman method [21]. This interviewing technique asks questions about the participants' history, perspectives, and reflections. The Seidman method provides a holistic understanding of the participants' perspectives and lived experiences. This study's semi-structured interview guide had three sections: focused history of climbing (e.g., "Please start by telling us about your current engagement with indoor climbing"); perceptions, insights, and experiences related to physical, mental, and social well-being (e.g., "In what way do you feel that indoor climbing helped you deal with stress?") and reflection and meaning of climbing. The full interview script is attached in the S1 Appendix.

### Data analysis

All interviews were recorded and transcribed. The qualitative results are presented as the anonymized individual details, excluding personally identifiable information of individual participants. Interviews were analyzed thematically. The first

**Table 1. Sociodemographic characteristics of participants.**

| Characteristics | N | % | Mean | SD |
|---|---|---|---|---|
| **Age** | | | 20.87 | 2.15 |
| 18 | 3 | 10 | | |
| 19 | 9 | 30 | | |
| 20 | 2 | 6.67 | | |
| 21 | 5 | 16.67 | | |
| 22 | 3 | 10 | | |
| 23 | 4 | 13.33 | | |
| 24 | 2 | 6.67 | | |
| 25 | 2 | 6.67 | | |
| **Gender** | | | | |
| Male | 14 | 46.67 | | |
| Female | 15 | 50 | | |
| Non-Binary/transgender | 1 | 3.33 | | |
| **Nativity** | | | | |
| US born American | 29 | 96.67 | | |
| Permanent resident (green card) | 1 | 3.33 | | |
| **Race/ethnicity** | | | | |
| Asian | 1 | 3.33 | | |
| White or European | 24 | 80 | | |
| Other | 5 | 16.67 | | |
| **Student Status** | | | | |
| Undergrad at University of Utah | 20 | 66.67 | | |
| Graduate at University of Utah | 1 | 3.33 | | |
| Undergrad at other university | 1 | 3.33 | | |
| Graduate at other university | 1 | 3.33 | | |
| Not a student and working | 7 | 23.33 | | |
| In-state students | | | | |
| No | 9 | 30 | | |
| Maybe | 14 | 46.67 | | |
| Not Applicable | 6 | 20 | | |
| **School year** | | | | |
| Freshman | 4 | 13.33 | | |
| Sophomore | 8 | 26.67 | | |
| Junior | 4 | 13.33 | | |
| Senior | 5 | 16.67 | | |
| Other | 2 | 6.67 | | |
| Not Applicable | 6 | 20 | | |
| **Living arrangement** | | | | |
| Alone | 3 | 10 | | |
| With family or relative(s) | 8 | 26.67 | | |
| With friend(s)/roommate(s) | 11 | 36.67 | | |
| Dormitory/resident hall/fraternity/sorority | 5 | 16.67 | | |
| Other | 3 | 10 | | |

stage of data analysis used a two-level thematic analysis approach. This stage used a table that had themes, subthemes, and quotes that fell within each theme and subtheme. Stage 2 was done to combine the tables made in Stage 1. Lastly, a codebook was made to organize themes, subthemes, and operationalization. Table 2 presents major themes found during data analysis, with respective explanation and quotes. To ensure credibility, transferability, and dependability, the interviewers were also climbers who understood the climbing community well and described details of climbing communities and procedures. For coder reliability, two coders coded separately while a third coder checked inter-rater reliability.

## Results

Fig 1 illustrates a conceptual model of the research findings.

### Focused history

Over half of participants climbed three or more times per week. One participant said, "I climb three days a week. Um, kind of like a two-hour session." Availability and social influence were the two primary factors responsible for how often participants climbed. Participants expressed that they would ideally climb more than they currently do, but their schedules cause time constraints. One participant explained, "I usually go…once a week. I would like to be going twice, but…my schedule [is] kind of odd because I work nights." Participants' engagement with climbing stemmed largely from their current social groups, especially for students: "I started doing it just because all my friends did it, [and] I kind of had fun." Participants who were students started climbing because their friends climbed, whereas those who were not students found friends through climbing.

There were two main reasons that participants began indoor climbing. One reason was that their family introduced climbing to them. One climber said, "my dad's been climbing my whole life, so I started climbing when I was in, I wanna say, fifth grade or fourth." The second reason was that they got known about climbing through online connections, including dating apps: "I actually went to Bumble Friends and I put, like, I want…a climbing partner."

Participants' climbing evolution over the years allowed for commitment to the sport:

"When I started, I was kind of bad at it. I was really scared of heights and falling off the wall and then kind of as I started doing it more and hanging out with people that did it a lot more, I started to get more comfortable with it, and started to go more frequently."

### Perceptions, insights, and experiences

Climbing is an exercise that participants felt encouraged feelings of a more balanced life overall. "It motivates me to actually, like, go and work out. It motivates me to eat healthy and then, uh, [psychologically, it] benefits [me]. It's a very good… hobby."

**Social life and social network.**  Three themes emerged from this subsection. First, indoor climbing created a space for new friendships and social connections. One interviewee answered, "I've met a lot of friends through climbing, and I've been able to hang out with, um, my current friends and family members more." Another participant explained, "...most of my friends, I met [through climbing]." Climbers both met their current friends through climbing and grew closer to their current friends by using their leisure time to climb together.

Second, indoor climbing promoted an increased sense of shared identity and community. One participant said, "The climbing community [in particular] is super active and social." Another participant explained that when they lived in another state, people at the gyms there kept to themselves. The other climbers at the gym did not spark conversations with people they did not already know. But in the current state, people were welcoming and inviting. Furthermore, one participant, who was an international student, explained that indoor climbing helped them feel like belonged in the community.

Table 2. *Codebook.*

| Primary Themes | Definition of theme | Representative quote |
|---|---|---|
| 3+ times/week | Participant climbs indoors three times or more per week | "I climb three days a week. Um, kind of like a two hour session of climbing and then two of the three days I'd weight lift for like an hour and a half before." (26) |
| Other commitments interfere | Participant expresses they'd ideally climb more than they currently do per week, but their schedules cause time conflicts. | I usually go I would say once a week. I would like to be going twice, but just with my schedule being kind of odd because I work nights. Um- it is not really possible for me to go sometimes twice a week." (2) |
| Social influences on frequency | Participant's engagement in climbing associated with their friends climbing | "I started climbing about a little bit over a year ago and I'd say I probably started doing it just because all my friends did it so I kind of had fun but I feel like it's a really good way to socialize, especially like with bouldering." (18) |
| Family influenced onset | Participant started climbing due to the influence of a family member | "My dad's been climbing my whole life, so I started climbing when I was in I wanna say 5th grade or 4th." (14) |
| Keep showing up | Participant's climbing evolution over the years allowed for commitment to the sport | "When I started, I was kind of bad at it. I was really scared of heights and falling off the wall and then kind of as I started doing it more and hanging out with people that did it a lot more, I started to get more comfortable with it, and started to go more frequently." (22) |
| Online connections | Participant found climbing partners online | "I actually went onto Bumble like friends and I put like I want like a climbing partner at the Front." (17) |
| Self-confidence | Climbing has benefitted participant's perceived self-worth | "Adrenaline or sense of satisfaction when you do when you accomplish something that you didn't think you could." (29) |
| Overall physical strength | Climbing utilizes many muscles that contribute to building strength | "One of my favorite things about climbing is, you never know like what like muscles you're gonna use. So it's really like a full body workout." (4) |
| Primary Themes | Definition of them | Representative quote |
| Balance | Climbing is an exercise participants feel encourages other life balances | "It motivates me to actually like go and work out, it motivates me to eat healthy and then uh psychological benefits it's a very good like hobby." (11) |
| Making friends | Climbing creates space for new connections | "Most of my friends I met on campus. And just like in general, has been like in the climbing gym, or just outdoor climbing just in the climbing scene in general" (30) |
| Community | Climbing brings sense of shared community | "Particularly, the climbing community is super active and social." (23) |
| Belayer/climber relationship | Partner rope climbing brings a fun and close social dynamic | "When you climb you have to trust those people around you cause it can be dangerous so it's like the outlying ability to trust other people and then build really close relationships with people who I can trust and I know like can be there for me." (14) |
| Difficulties with the climbing social scene | Connections made at climbing gyms have not expanded past the gym | "I did make friends, but sometimes, I feel like they're not, like, friends-friends." (33) |
| Coping with stress | Participant utilizes climbing as an outlet for pent up emotion | "I was just juggling a lot of different things and it felt really good to, like, come into the gym and put all that energy into climbing hard" (26) |
| Mindfulness | Climbing allows participant to stay in the present moment | "It kind of helps me take my mind off of the things that are stressing me out. It just helps me to just, I don't know, kind of just get grounded." (27) |
| Injuries create stress | Climbing-related injury directed into stress towards climbing | "Sometimes it actually contributes to my stress because I've had so many injuries" (21) |
| Problem-solving | Participant is challenged to creative solutions when climbing routes | "I think it's kind of cool to see that like just because you don't do something the first time, like, if you keep working at it, you'll get better at it and you will eventually do good." (22) |
| Primary Theme | Definition of theme | Representative quote |
| Routine | Climbing brings sense of normal and routine | "Having something where no matter where, like how I'm feeling, I can go and then have it literally like two minutes away with good people." (5) |
| Healthier lifestyle | Engaging with climbing has positively influenced other health aspects | "It's gotten me to work out, it's gotten me to eat better, I used to be pretty weak and now I'm moderately strong." (11) |
| Expansion to other sports | Climbing motivated participant to try other physical activities to complement climbing | "Recently I've had like a lot of goals for climbing so it also encourages me to like do other physical activity that I know is like gonna benefit my climbing performance." (35) |
| Sleep benefits | Physical activity of climbing helps sleeping at night | "I think it helps like my sleep too, because if you're active during the day, you sleep better at night." (3) |

*(Continued)*

Table 2. (Continued)

| Primary Themes | Definition of theme | Representative quote |
|---|---|---|
| **Financial barriers** | Pricing for a membership at a climbing gym may prevent someone from accessing | "I mean 85 bucks a month…usually the main reasons I would stop a membership is kind of financial reasons to save the money." (31) |
| **Male-dominated sport** | Climbing as a marginalized gender identity in a male-dominant space may feel uncomfortable | "As a woman, it's kind of really intimidating to, to go to the gym and see, like, just mostly men. Any that's the main reason why I didn't do it for a while. Like that's why I don't use the student rec center as well." (16) |
| **Overuse injuries** | Climbing is a physically demanding sport that can result in injury if not approached with precaution | "There's a risk of injury at all times. Like I partially tore two of my pulleys last year so that was not fun." (1) |
| **Mental frustration** | Can put unrealistic pressures on oneself to produce results in climbing | "I got really frustrated that it- it took me forever, um, to- to like, get progress." (33) |
| **Reframe exercise** | Climbing allowed participant to view physical movement in a positive light | "It's made me like reframe uh- you know like physical movement like it doesn't have to be like – it can be fun, and it doesn't – it can be something that I do for fun in my free time. It doesn't have to be like this high intensity workout that I am doing." (2) |
| **Primary Theme** | **Definition of theme** | **Representative quote** |
| **Utilizing leisure time** | Climbing influences more productive leisure time | "Choosing to go for a climb instead of just laying around scrolling on my phone or whatnot, is always a good thing." (7) |
| **Body image** | Climbing has improved perception of one's body | "I feel like it helps me work out a lot which helps me stay fit, which means I'm more confident and like, feel confident about how I look." (31) |
| **Outdoor climbing** | Indoor climbing can lead the way to trying outdoor | "Outdoor climbing puts you like in close contact with nature, there's like a certain feeling of risk that you get outdoors that you don't get in a gym that I really enjoy." (11) |
| **Climbing is dangerous** | There are risks to climbing that need to be taken seriously | "Climbing is a very dangerous sport. And so that is, I don't know. That's something to keep in mind is double checking and safety." (4) |

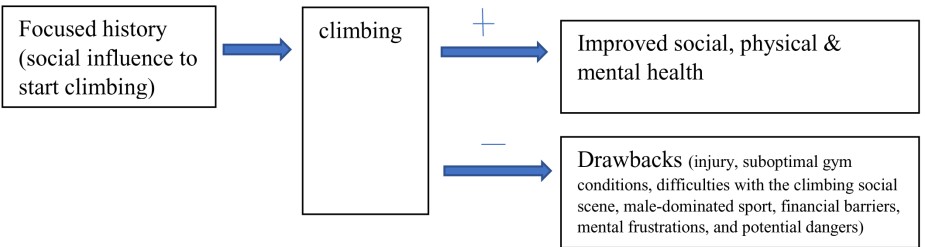

**Fig 1. A conceptual model illustrating the research findings.**

Third, there was a belayer-climber relationship where the climbing partnership brought a fun and close social dynamic to the sport:

"When you climb, you have to trust those people around you 'cause it can be dangerous. So, it's like the outlying ability to trust other people and then build really close relationships with people who I can trust, and I know, like, can be there for me."

The participant expressed that they had been able to build trust and connection through the sport, particularly through top rope climbing, which is the type of climbing that involves a partner.

**Coping with stress.** Many of the participants felt that indoor climbing acted as a form of mindfulness, provided communal stress relief, and helped students specifically to decrease their stress levels. One person put it into words: "It's like an adult playground...it looks fun...your mind is just focused on the wall. You're not thinking about all the stresses of

life." Another participant explained that "it kind of helps me take my mind off of the things that are stressing me out. It just helps me to just, I don't know, kind of just get grounded."

Participants also felt that indoor climbing was a form of communal stress relief. One student explained that they and their friends from school would go to their local climbing gym and rotate between exercising and studying. She expressed that she could notice a shift in energy after climbing. Before, she said, they were all stressed, but when they climbed, they experienced "mental reset." Another participant explained that climbing helped them release stressful, pent-up emotions: "I was just juggling a lot of different things and it felt really good to, like, come into the gym and put all that energy into climbing hard."

We also saw from our interviews that indoor climbing helped college students positively cope with stress. "It's definitely become a healthy outlet for me, like, if I had a tough day at school or I'm stressed with classes," said one of our student participants. The above examples emphasize this theme.

**Mental health.** Indoor climbing promoted a sense of regulation and routine. It brought a sense of normalcy into one's life. It's important, one student said, to have "Something where no matter [what], like how I'm feeling, I can go [climb]." This same participant also explained that they know when they climb, they are "surrounded by good people", and that also helped them to have a stronger sense of mental well-being.

In addition, indoor climbing promoted confidence and self-discovery. It benefited participants' perceived self-worth. One participant said they got a surge of "adrenaline and sense of satisfaction when [they] accomplished something that [they] didn't think [they] could." Another student said, "...for anything else in the past, it's kind of, [like,] my parents led me into that field…But climbing, it's something that I ch- I chose myself. So I don't know, I like indoor climbing the most out of all the sports."

Problem solving also came up as a dominant theme. The climber is challenged to create solutions when climbing a route. One participant remarked, "I think it's kind of cool to see that like, just because you don't do something the first time, like, if you keep working at it, you'll get better at it and you will eventually do good."

**Physical health.** Participants strongly agreed that indoor climbing helped their physical health. Specifically, five themes appeared. First, indoor climbing built overall physical strength. "One of my favorite things about climbing is [that] you never know what muscles you're gonna use. So, it's really like a full body workout," said one of the participants. Second, indoor climbing facilitated living a healthier lifestyle. Many participants spoke about how engaging with climbing had positively influenced other health aspects for them. For example, one student participant expressed, "...it's gotten me to work out. It's gotten me to eat better. I used to be pretty weak and now I'm moderately strong." Third, indoor climbing is unique in that it facilitated a balance between intensity and intentional rest. One participant shared that when they bike it's "...just…headphones in and I'm just breathing really heavily for a long time...whereas climbing is like... And especially if I'm bouldering, I'm trying really hard, and then I'm resting for a few minutes, and then that's my chance to kinda chat with people and chill." Fourth, indoor climbing has helped participants expand into other sports by motivating them to try other physical activities to complement climbing. "Recently, I've had a lot of goals for climbing so it also encourages me to do other physical activities that I know are gonna benefit my climbing performance," one participant said. For some participants, indoor climbing served as a gateway to outdoor climbing. One participant explained that "...outdoor climbing puts you in close contact with nature. There's like, a certain feeling of risk that you get outdoors that you don't get in a gym that I really enjoy." Fifth, some participants felt indoor climbing helped them sleep better at night: "I think it helps my sleep too because if you're active during the day, you sleep better at night."

**Drawbacks and issues.** Though almost all participants felt that indoor climbing had a net positive effect on their well-being and felt that it benefited their lives, there were some noted drawbacks, including: risk of personal injury or danger, dissatisfaction with suboptimal gym conditions, difficulties with the climbing social scene and its male-dominated dynamic, financial barriers, mental frustration.

Injury was one of the most prominent themes. One participant said, "...there is a chance of, like, some repetitive strain injuries in, like, finger tendons which is just really bad news." Injuries can also create stress. One participant explained that climbing-related injuries caused stress in her life: "Sometimes [climbing] actually contributes to my stress because I've had so many injuries." Climbing is a physically demanding sport that can result in injury, especially overuse injuries, if not approached with caution. "There is a risk of injury at all times. Like, I partially tore two of my [pully muscles] last year so that was not fun," said one participant.

Some participants spoke about the importance of gym amenities to their overall experience, such as a sauna or the availability of lotion and hair ties. One person said that having the sauna in the gym made going to exercise a longer and more relaxing experience. Another participant explained that they determine which climbing gym to go to based on the amenities offered. But not all climbing gyms have such amenities.

Gender disparities and financial barriers are the issues emerged from the interviews. Climbing is a male-dominated sport. One person said that "...as a woman, it's kind of really intimidating to, to go to the gym and see, like, just mostly men. That's the main reason why I didn't do it for a while. Like, that's why I don't use the student rec center as well." Indoor climbing can be expensive. One student said, "I mean, eighty-five bucks a month…usually the main reasons I would stop a membership is kind of financial reasons to save the money."

Climbing can also lead to mental frustration—in contrast to the mental health benefits mentioned previously. The sport can put unrealistic pressures on oneself to produce results while climbing. As explained by one participant, "I got really frustrated that it, it took me forever, um, to, to like, [make] progress."

Lastly, while indoor climbing is less risky than outdoor climbing, there are risks that need to be taken seriously. "Climbing is a very dangerous sport. And so that is, I don't know. That's something to keep in mind is double-checking and safety." Even with indoor climbing, if you are not taking the right precautions, the effects can be quite harmful.

### Reflection and meaning

When asking participants how indoor climbing has changed their sense of leisure time, they claimed climbing helped them to use their leisure time in a way that aligned with how they want to live their life. For example, one person said, "...choosing to go for a climb instead of just laying around scrolling on my phone or whatnot, is always a good thing." To conclude each interview, participants were asked if there was anything else they would like to say. There were some noted final themes.

Indoor climbing completely changed some participants' sense of self and body image: "I've been kind of insecure about my body for a long time, but climbing has been one of the ways to get me out of that and to feel good about it." Another respondent said, "I feel like it helps me work out a lot which helps me stay fit, which means I'm more confident and like, feel confident about how I look." Our participants felt deeply passionate about this theme. Another participant went on to say that "I think climbing could be almost life saving for a lot of people. Like, it's more than just a sport."

Indoor climbing has helped participants to reframe what exercise can look like. "It's made me, like, reframe uh—you know—like, physical movement like…it can be fun, and…it can be something that I do for fun in my free time. It doesn't have to be like, this high intensity workout that I am doing." Indoor climbing has allowed people to view physical movement in a positive light. As one participant put it, "the best climber is the one having the most fun."

### Discussion

In the exploration of indoor climbing and the well-being of young adults, this study has illuminated several key aspects that contribute to its positive impacts, such as improved physical health, mental health (e.g., stress release, self-esteem, body image), and the development of a supportive community.

Indoor climbing helped participants' physical health in a multitude of ways. Specifically, participants spoke of how they became more mindful of their sleeping and eating habits. Because of indoor climbing, many participants were also

motivated to take up other forms of physical activity like outdoor climbing, weightlifting, and yoga. Further, indoor climbing was able to help many participants feel more balanced both on and off the climbing wall. Participants emphasized that they felt indoor climbing was more conducive to taking intentional periods of rest than other sports they were involved in. Rest is important to restore energy, or glycogen, and helps prevent injury and chronic muscle pain [22]. Indoor climbing has the potential to support incorporating intentional rest into physical activity and promote a health-conscious lifestyle.

The results of this study indicate that indoor climbing can be a beneficial method for coping with stress to cope with stress. High levels of stress can lead to increased college student drop-out rates and reduce motivation [23]. Furthermore, most college students tend to cope with stress in unhealthy ways, "like staying alone and sleeping too much" [24]. The findings of this study show that indoor climbing has the potential to help young adults cope with stress in a healthy way, supporting both healthy sleep habits and social connectedness.

The transformative impact of climbing on self-esteem and body image was a recurring theme in the responses. Participants noted improvements in body confidence and overall self-esteem as direct outcomes of their involvement in climbing. The results are consistent with previous studies that found a positive association between sports and positive body image [25]. These personal testimonies highlight the sport's potential as not just a recreational activity but as a potentially therapeutic one [26].

The impact of indoor climbing on participants' use of leisure time also emerged as a significant finding. Many respondents reported that climbing encouraged them to make more intentional and fulfilling use of their free time, choosing active engagement over passive activities like scrolling through social media. This shift not only enhances physical health but also contributes to better mental health and a more satisfying lifestyle. Previous research indicated that young adults tend to perceive physical activity as a responsibility rather than leisure while they believe that home-based activities (e.g., computer games) are leisure [27]. Indoor climbing has the potential to promote physical activity as a form of leisure to young adults.

While there are positive impacts of indoor climbing, there are some challenges, including gender disparities and financial cost, that can make it difficult to get involved in the sport. The 2023 Outdoor Participation Trends Report showed 58% men and 42% women among climbers [28]. It is important to provide support to those who may need extra attention to be comfortable in the climbing community.

Furthermore, the financial aspect of indoor climbing, marked by high membership fees at many climbing gyms, poses an additional barrier. Financial barriers are a serious problem to promote sports among youth [29]. The membership of a climbing gym costs approximately $80 per month [1]. This is especially significant for college students and other vulnerable populations who may find these costs prohibitive, thus limiting their ability to engage with the sport despite its benefits. To promote indoor climbing among young adults, it is essential to develop financial support mechanisms.

Although this study captured the insights of the college-age population, which could have policy implications for health promotion initiatives surrounding indoor climbing to decrease stress among students, there are limitations. This study only reflects young adult indoor climbers—mostly students— in a state that has a strong climbing community. The results may not be applicable to other states where climbing is not as popular. Future research should have study participants from different regions. Further, our demographic data represents mostly non-Hispanic White participants. The city where this study was carried out has a predominantly white population (71%). Further research should be conducted to examine racial and ethnic disparities among indoor climbers. Moreover, since the participants are a convenience sample who regularly climbed, there is a possibility that the participants were more likely to have had positive attitudes toward indoor climbing compared to those who do not climb.

## Conclusion

This study is expected to help develop effective health promotion programs using indoor climbing for college students and young adults by having explored indoor climbing and physical, mental and social well-being among college-aged

young adults based on interviews with indoor climbers. Social connection is an important component to ensure health and well-being of college students based on the results of this study and social influence theory. College campuses could create larger indoor climbing spaces or partner with local climbing gyms to encourage students to climb to help increase student well-being and success. Indoor climbing could help address the mental health crisis on college campuses without burdening the already overwhelmed student counseling centers. Further, this study may help healthcare providers recognize the benefits of climbing on the health and well-being of young adults. Community health educators could promote indoor climbing as a beneficial form of physical activity that could enhance physical, mental, and social well-being to support young adults.

These findings collectively emphasize the multifaceted benefits of indoor climbing, while also acknowledging the socio-economic and cultural hurdles that need addressing to make these benefits more universally accessible. Future research should focus on gender, racial/ethnic, and socio-economic disparities to develop more inclusive and accessible indoor climbing communities. In addition, future studies using quantitative methods would be beneficial to further examine the health benefits of indoor climbing for young adults.

## Supporting information

**S1 Appendix. Interview script.**
(DOCX)

## Author contributions

**Conceptualization:** Jeff Rose, Max E. Coleman, Timothy A. Brusseau Jr, Kathy Franchek-Roa, Akiko Kamimura.

**Data curation:** Elsa C. Osborne, Logan Reeves, Kelli Spear, Akiko Kamimura.

**Formal analysis:** Elsa C. Osborne, Logan Reeves, Kelli Spear, Akiko Kamimura.

**Funding acquisition:** Jeff Rose, Kathy Franchek-Roa, Akiko Kamimura.

**Investigation:** Akiko Kamimura.

**Methodology:** Akiko Kamimura.

**Project administration:** Elsa C. Osborne, Akiko Kamimura.

**Resources:** Akiko Kamimura.

**Supervision:** Akiko Kamimura.

**Writing – original draft:** Elsa C. Osborne.

**Writing – review & editing:** Elsa C. Osborne, Jeff Rose, Logan Reeves, Kelli Spear, Max E. Coleman, Timothy A. Brusseau Jr, Kathy Franchek-Roa, Akiko Kamimura.

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
