## [Decision Letter · Decision Letter 0]

23 Dec 2024

PONE-D-24-41965Indoor Climbing and Well-being of Young Adults:

Perspectives among indoor climbersPLOS ONE

Dear Dr. Kamimura,

Thank you for submitting your manuscript to PLOS ONE. After careful consideration, we feel that it has merit but does not fully meet PLOS ONE’s publication criteria as it currently stands. Therefore, we invite you to submit a revised version of the manuscript that addresses the points raised during the review process. **Please revise the manuscript according to the reviewers' comments and re-submit the revised version of the manuscript on due time.**

We look forward to receiving your revised manuscript.

Kind regards,

Rasool Abedanzadeh, Ph.D

Academic Editor

PLOS ONE

**Journal Requirements:**

University of Utah

This research was supported by 1U4U Innovation Funding and the Undergraduate Research Opportunity Program at the University of Utah. The earlier version of this manuscript was the first author’s honor’s thesis. 

University of Utah

4. We noted in your submission details that a portion of your manuscript may have been presented or published elsewhere. The earlier version of this manuscript was the first author's honor's thesis. Publishing theses/dissertation in a peer-reviewed journal is usually not considered as dual publication. Please clarify whether this [conference proceeding or publication] was peer-reviewed and formally published. If this work was previously peer-reviewed and published, in the cover letter please provide the reason that this work does not constitute dual publication and should be included in the current manuscript.

5. Please provide a complete Data Availability Statement in the submission form, ensuring you include all necessary access information or a reason for why you are unable to make your data freely accessible. If your research concerns only data provided within your submission, please write "All data are in the manuscript and/or supporting information files" as your Data Availability Statement.

Reviewers' comments:

Reviewer's Responses to Questions

**Comments to the Author**

1. Is the manuscript technically sound, and do the data support the conclusions?

Reviewer #1: Yes

Reviewer #2: Partly

2. Has the statistical analysis been performed appropriately and rigorously? 

Reviewer #1: Yes

Reviewer #2: No

3. Have the authors made all data underlying the findings in their manuscript fully available?

Reviewer #1: No

Reviewer #2: No

4. Is the manuscript presented in an intelligible fashion and written in standard English?

Reviewer #1: Yes

Reviewer #2: Yes

5. Review Comments to the Author

**Reviewer #1:**  As you know, different parts of the article need corrections, and in order to edit and complete them, you need to read and correct the mentioned items in different parts. Different sections include abstract. Introduction. method Results. discussion

**Reviewer #2:**  Abstract:

Please ensure the following points are addressed in the abstract:

The introduction is too lengthy; it would be better if it is shortened.

The objective related to the title should be clearly stated.

The type of research and the research method should be mentioned.

The coding approach and the method of analyzing the data from the interviews should be outlined.

The conclusion should correspond with the findings.

Introduction:

The introduction requires rewriting.

Please revise the introduction to transition from a thesis style to an article format.

Avoid including unnecessary headings in the introduction.

Ensure coherence and consistency throughout the content.

Present the introduction as a single section, starting with the main variables of the research, discussing the challenges and the need for the study, referencing related studies and their limitations, and finally stating the main objective or hypothesis.

Research Methodology:

Provide a thorough explanation of the research method and type at the beginning of this section. Follow this with details regarding the participants and the sampling method used, and conclude with the execution and data analysis process. Write this section in line with article standards, avoiding a thesis-like style.

Clarify whether coding was conducted during the interviews and explain how saturation was achieved.

Considering that your sample comprised students, justify the use of a snowball sampling method.

Include a detailed description of the interview questions in the methodology section.

Explain how credibility, transferability, and dependability—key aspects of qualitative studies—were addressed.

Ensure inter-rater reliability (coder reliability) is established, and report the results.

Results:

Given the length of this section, avoid unnecessary details and focus on the main and subcategories.

Present the main and subcategories, along with the extracted concepts, in tables.

If coding was performed during the interviews, include a table reporting the interview codes.

To aid understanding, provide a conceptual model illustrating the research findings.

Discussion:

Revise the discussion to align with the research findings.

Start by presenting the research findings, compare them with previous studies, and conclude with a comprehensive summary.

Mention the limitations of the research and offer practical suggestions

6. PLOS authors have the option to publish the peer review history of their article (what does this mean? ). If published, this will include your full peer review and any attached files.

**Do you want your identity to be public for this peer review?** For information about this choice, including consent withdrawal, please see our Privacy Policy .

Reviewer #1: No

Reviewer #2: No

---

## [Author Response · Author response to Decision Letter 1]

11 Feb 2025

Response to the reviewers

Reviewer 1

As you know, different parts of the article need corrections, and in order to edit and complete them, you need to read and correct the mentioned items in different parts. Different sections include abstract. Introduction. method Results. Discussion

Response: Thank you for your comments. We made revisions below.

Reviewer – one of the attachments

Abstract: To summarize, refer to the statistical method

Response: This is a qualitative study and data analysis was not based on statistical analysis. We clarified the methodological approach.

Introduction: Strengthen theoretical foundations (both positive and negative research)

Response: We revised the paragraph about the theory to make the theoretical foundation clearer.

Methods: It would have been better if people were selected from different regions

Response: Great point. We added it to the limitation section.

Data Analysis: In this section, refer to the data analysis in thematic form

Response: We presented it in Table 2 and made it clearer in the method section.

Discussion: In the section rejecting or confirming proposals, strengthen the theoretical foundations with the results of your work

Response: We included that point in conclusion.

Reviewer 2

Review of the Article: " Indoor Climbing and Well-being of Young Adults: Perspectives among indoor climbers"

Dear Authors,

To enhance the quality of this article, I kindly suggest the following revisions:

Abstract:

Please ensure the following points are addressed in the abstract:

The introduction is too lengthy; it would be better if it is shorten.

Response: We shortened it.

The objective related to the title should be clearly stated.

Response: Done

The type of research and the research method should be mentioned.

Response: Done

The coding approach and the method of analyzing the data from the interviews should be outlined.

Response: Done

The conclusion should correspond with the findings. Response: We revised the conclusion.

Introduction:

The introduction requires rewriting.

Response: We revised the introduction.

Please revise the introduction to transition from a thesis style to an article format.

Response: We revised the introduction.

Avoid including unnecessary headings in the introduction.

Response: We dropped the sub-headings.

Ensure coherence and consistency throughout the content.

Present the introduction as a single section, starting with the main variables of the research, discussing the challenges and the need for the study, referencing related studies and their limitations, and finally stating the main objective or hypothesis.

Response: We made the introduction to be a single section and followed the structure that you suggested. However, because this is a qualitative study, there is no hypothesis to present.

Research Methodology:

Provide a thorough explanation of the research method and type at the beginning of this section. Follow this with details regarding the participants and the sampling method used, and conclude with the execution and data analysis process. Write this section in line with article standards, avoiding a thesis-like style. Response: Thank you for your comment

Response: Thank you for your comments. We revised the method section.

Clarify whether coding was conducted during the interviews and explain how saturation was achieved. Response: Coding was conducted AFTER the interviews.

Considering that your sample comprised students, justify the use of a snowball sampling method.

Response: We justified the snowball sampling method.

Include a detailed description of the interview questions in the methodology section.

Response: We included more details about the in the description of the interview questions.

Explain how credibility, transferability, and dependability—key aspects of qualitative studies—were addressed.

Response: We added the information.

Ensure inter-rater reliability (coder reliability) is established, and report the results.

Response: We added the information.

Results:

Given the length of this section, avoid unnecessary details and focus on the main and subcategories.

Response:

Present the main and subcategories, along with the extracted concepts, in tables.

Response: It’s presented in Table 2.

If coding was performed during the interviews, include a table reporting the interview codes.

Response: Coding was not performed during the interviews.

To aid understanding, provide a conceptual model illustrating the research findings.

Response: We added Figure 1 that illustrates a conceptual model of the findings.

Discussion:

Revise the discussion to align with the research findings.

Start by presenting the research findings, compare them with previous studies, and conclude with a comprehensive summary.

Response: We re-organized the discussion section.

Mention the limitations of the research and offer practical suggestions.

Response: We added practical suggestions to the limitation section.

Final Remarks:

Your research topic is very interesting. However, more effort is needed to strengthen the writing in all sections to improve the overall quality and presentation of the article.

Response: Thank you for your comments. We hope that the paper has improved.

---

## [Decision Letter · Decision Letter 1]

10 Mar 2025

Indoor Climbing and Well-being of Young Adults:

Perspectives among indoor climbers

PONE-D-24-41965R1

Dear Dr. Kamimura,

We’re pleased to inform you that your manuscript has been judged scientifically suitable for publication and will be formally accepted for publication once it meets all outstanding technical requirements.

Kind regards,

Rasool Abedanzadeh, Ph.D

Academic Editor

PLOS ONE

Additional Editor Comments (optional):

Reviewers' comments:

Reviewer's Responses to Questions

**Comments to the Author**

1. If the authors have adequately addressed your comments raised in a previous round of review and you feel that this manuscript is now acceptable for publication, you may indicate that here to bypass the “Comments to the Author” section, enter your conflict of interest statement in the “Confidential to Editor” section, and submit your "Accept" recommendation.

Reviewer #1: All comments have been addressed

Reviewer #2: All comments have been addressed

2. Is the manuscript technically sound, and do the data support the conclusions?

Reviewer #1: Yes

Reviewer #2: Yes

3. Has the statistical analysis been performed appropriately and rigorously? 

Reviewer #1: Yes

Reviewer #2: Yes

4. Have the authors made all data underlying the findings in their manuscript fully available?

Reviewer #1: Yes

Reviewer #2: Yes

5. Is the manuscript presented in an intelligible fashion and written in standard English?

Reviewer #1: No

Reviewer #2: Yes

6. Review Comments to the Author

Reviewer #1: (No Response)

Reviewer #2: (No Response)

7. PLOS authors have the option to publish the peer review history of their article (what does this mean? ). If published, this will include your full peer review and any attached files.

**Do you want your identity to be public for this peer review?** For information about this choice, including consent withdrawal, please see our Privacy Policy .

Reviewer #1: No

Reviewer #2: No

---

## [Editor Report · Acceptance letter]

PONE-D-24-41965R1

PLOS ONE

Dear Dr. Kamimura,

I'm pleased to inform you that your manuscript has been deemed suitable for publication in PLOS ONE. Congratulations! Your manuscript is now being handed over to our production team.

Kind regards,

on behalf of

Dr. Rasool Abedanzadeh

Academic Editor

PLOS ONE